# Adaptive Dense Reward: Understanding the Gap Between Action and Reward Space in Alignment

## Abstract

Reinforcement Learning from Human Feedback (RLHF) has proven highly effective in aligning Large Language Models (LLMs) with human preferences. However, the original RLHF typically optimizes under an overall reward, which can lead to a suboptimal learning process. This limitation stems from RLHF's lack of awareness regarding which specific tokens should be reinforced or suppressed. Moreover, conflicts in supervision can arise, for instance, when a chosen response includes erroneous tokens, while a rejected response contains accurate elements. To rectify these shortcomings, increasing dense reward methods, such as step-wise and token-wise RLHF, have been proposed. However, these existing methods are limited to specific tasks (like mathematics). In this paper, we propose the "Adaptive Message-wise RLHF" method, which robustly applies to various tasks. By defining pivot tokens as key indicators, our approach adaptively identifies essential information and converts sequence-level supervision into fine-grained, subsequence-level supervision. This aligns the density of rewards and action spaces more closely with the information density of the input. Experiments demonstrate that our method can be integrated into various training methods, significantly mitigating hallucinations and catastrophic forgetting problems, while outperforming other methods on multiple evaluation metrics. Our method improves the success rate on adversarial samples by 10% compared to the sample-wise approach, and achieves a 1.3% improvement on evaluation benchmarks such as MMLU, GSM8K, HumanEval, etc.

## 1 Introduction

In recent years, generative AI models have made significant achievements, with preference alignment by reinforcement learning playing an essential role in this progress (Ouyang et al., 2022; Touvron et al., 2023; Rafailov et al., 2024; Dubey et al., 2024; Yang et al., 2024a; OpenAI et al., 2024). These methods, such as reinforcement learning from human feedback (RLHF), mostly involve sampling from the policy, obtaining rewards, enhancing the logits of positive samples according to rewards, and reducing the logits of negative samples. However, dealing with sparse rewards is challenging in reinforcement learning: can large language models effectively learn whether each token is correct through sparse reward signals, like scoring on a whole sample? This paper presents a theoretical analysis and experimental investigation into this issue, aiming to identify the most effective strategy of reward signals for guiding the utilization of reward models in preference alignment by reinforcement learning.

The work of Rafailov et al. (2024) represents the autoregressive reward models using a contextual bandit framework. Studies by Radford et al. (2019) and Zhong et al. (2024) conceptualize the reward model as a token-level Markov Decision Process (MDP). These approaches illustrate that the reward model possesses fine-grained reward capabilities, but they do not provide the error equation between the reward signal generated by the model and the actual reward scores. Building on this foundation, our paper further quantifies the error in the reward signal. Our work indicates that this error mainly arises from the coarse-grained nature of the reward signal compared to the actual rewards and the inherent stochastic errors within the reward model itself. The coarse-grained reward signal can be optimized by reducing its granularity into a finer granularity, while the inherent stochastic errors can

be mitigated by using the overall reward of a longer sequence to represent the rewards of individual tokens or subsequences. This indicates that when assigning a reward signal to a sequence, it is important to use a fine-grained approach to provide different scores for its various parts. However, overly granular segmentation can increase the reward error. This demonstrates the need to align the density gap of the information within the sequence with the reward signal to reduce the total reward error, thereby improving the accuracy of the rewards (figure 1).

Leveraging the above theoretical framework, we further propose a method: "Adaptive Message-wise RLHF" shown in figure 3. This approach identifies key signals through rewards and generalized advantages during the generation process, allowing the model to adaptively partition samples. The resulting sub-sequences offer flexible control over gradient propagation through various methods. This adaptability in gradient management enhances the model's learning capabilities and reduces model hallucinations.

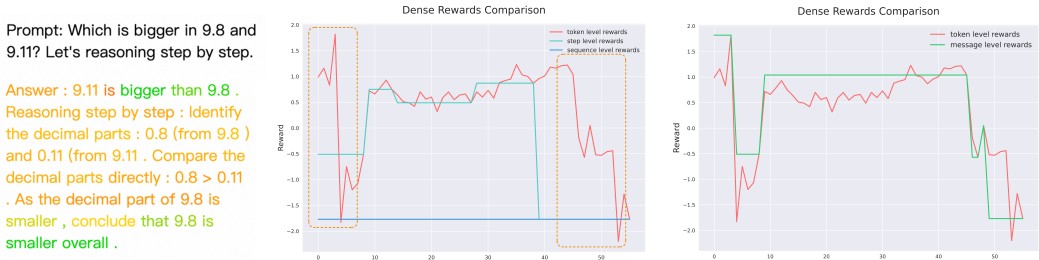

Figure 1: Comparison of different ways of reward signals. **Left**: Green represents low-reward tokens, while orange represents high-reward tokens. **Mid**: A comparison of token-wise, step-wise, and sequence-wise reward signals. Token-wise rewards exhibit significant fluctuations and high noise levels, leading to unstable training. The lines highlighted by the yellow dashed box shows that in the step-wise approach, tokens at the same step can have completely different rewards, yet they are all represented by the same reward score, which can lead to errors. **Right**: This image presents the message-wise reward method proposed in this paper. In autoregressive generative transformer, each token represents an action, The size of the vocabulary is the size of the action space. This method significantly separates tokens with different reward scores into distinct subsequences, thereby preventing the same subsequence from containing both positive and negative actions.

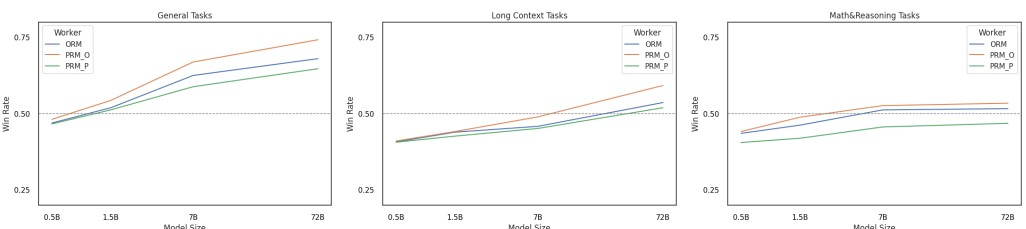

Figure 2: For general tasks, especially in low-information statement, long context(e.g., writing articles or RAG applications), step-wise supervision is significantly less accurate than sequence-wise supervision. Left and middle shows the performance of Outcome-supervised is better than Process-supervised for general tasks, even when well-trained PRMs(Appendix:A were used. ORM: Outcome supervised Reward Model. PRM:Process supervised Reward Model. $PRM_O$: Outcome supervised by PRMs. $PRM_P$: Process supervised by PRMs.

## 2 PRELIMINARIES

In this section, we present an overview of predominant RLHF methods. Additionally, we also define the symbols that will be consistently employed throughout this paper, as these notations will be integrated into our framework.

## 2.1 RL Environment

The Reinforcement Learning from Human Feedback (RLHF) paradigm can be formalized as a Markov Decision Process (MDP) with a dense reward structure derived from preference models. Traditionally, the Bradley-Terry model has been employed to estimate preferences between pairs of sequences:

$$\mathbb{P}(y_1 \succ y_2|x, y_1, y_2) = \sigma(r(x, y_1) - r(x, y_2)) \tag{1}$$

where $\sigma$ is the sigmoid function, $x$ is the input context, and $y_1, y_2$ are candidate responses. To refine this approach for token-level optimization, we decompose the reward function into individual token contributions:

$$\mathbb{P}(y_1 \succ y_2|x, y_1, y_2) = \sigma\left(\sum_{h=1}^{H} r(s_h^1, a_h^1) - \sum_{h=1}^{H} r(s_h^2, a_h^2)\right) \tag{2}$$

where $s_h^i$ and $a_h^i$ represent the state and action at step $h$ for sequence $i$, respectively. This decomposition allows the definition of both sequence-level and token-level reward functions:

$$r_s(x, y) = \log \pi^*(y|x), \quad r_t((x, y_{1:h-1}), y_h) = \log \pi^*(y_h|x, y_{1:h-1}). \tag{3}$$

Here, $r_s$ denotes the sequence-wise reward, while $r_t$ represents the token-wise reward. $\pi^*$ is the optimal policy derived from human preferences. This formulation enables a more granular approach to RLHF, allowing for token-by-token optimization. This effectively bridges the gap between preference learning and reinforcement learning, providing a dense reward signal that can guide policy improvement at a finer scale. The token-wise reward structure aligns with recent advancements in selective token methods (Yang et al., 2024b; Lin et al., 2024; Zeng et al., 2024a), which focus on optimizing the most relevant tokens. This synergy between dense reward modeling and selective token optimization presents a promising direction for improving the efficiency and effectiveness of RLHF in large language models.

**Process-supervised Reward Models**. Process-Supervised Reward Models (PRMs) were first introduced in Lightman et al. (2023b). This work proposes a method for training reward models that provides feedback by evaluating the correctness of each step in the solutions generated by the model. Unlike supervision based solely on the final answer, known as outcome supervision, process supervision offers explicit feedback for each step, allowing the model to learn to follow a reasoning process that is approved by humans. This approach simplifies the credit assignment task by providing more precise feedback and encourages the model to generate reasoning chains that align more closely with human expectations.

## 2.2 RL Methods

The evolution of Reinforcement Learning from Human Feedback (RLHF) has led to several methodological variants, each addressing specific aspects of the learning process. This section outlines key approaches in the RLHF paradigm.

### 2.2.1 Classical RLHF

The traditional RLHF objective function is formulated as:

$$\mathcal{L}_{\text{PPO}}(\theta) = \mathbb{E}_{x \sim D, y \sim \pi_\theta(y|x)} [r_\Phi(x, y)] - \beta D_{\text{KL}} [\pi_\theta(y|x) \parallel \pi_{\text{ref}}(y|x)], \tag{4}$$

where $r_\Phi(x, y)$ is the learned reward function, $\pi_\theta$ is the policy being optimized, $\pi_{ref}$ is a reference policy, and $\beta$ controls the strength of the KL-divergence regularization.

### 2.2.2 Direct Preference Optimization (DPO)

DPO reformulates RLHF as a preference learning problem:

$$P(y_1 > y_2|x) = \frac{\exp(r(x, y_1))}{\exp(r(x, y_1)) + \exp(r(x, y_2))}, \tag{5}$$

This leads to the DPO loss:

$$\mathcal{L}_{\text{DPO}}(\pi_\theta; \pi_{\text{ref}}) = -\mathbb{E}_{(x, y_w, y_l) \sim \mathcal{D}} \left[\log \sigma \left(\beta \log \frac{\pi_\theta(y_w \mid x)}{\pi_{\text{ref}}(y_w \mid x)} - \beta \log \frac{\pi_\theta(y_l \mid x)}{\pi_{\text{ref}}(y_l \mid x)}\right)\right], \tag{6}$$

where $y_w$ and $y_l$ represent winning and losing completions respectively.

### 2.2.3 REJECTION SAMPLING

An alternative approach uses rejection sampling, optimizing:

$$\mathcal{L}_{\text{Rejection Sampling}}(\theta) = -\mathbb{E}_{x, y_w \sim D} \left[ \log \pi_\theta(y_w | x) - \beta D_{\text{KL}}(\pi_\theta \parallel \pi_{\text{ref}}) \right], \tag{7}$$

This method directly optimizes the policy to generate preferred outputs while maintaining proximity to the reference policy.

### 2.3 ALIGNMENT

### 2.3.1 PREFERENCE ALIGNMENT

In the context of preference alignment, various methods such as Reinforcement Learning from Human Feedback (RLHF), Direct Preference Optimization (DPO), and Rejection Sampling can be unified under a common optimization framework. This framework involves a strategy $\pi_\theta(x)$ and a preference feedback function $F(x)$, which incorporates reward signals, preference differences, or rejection conditions. The generalized optimization objective can be formulated as:

$$\max_\theta \mathbb{E}_{x \sim \pi_\theta} [F(x)] \tag{8}$$

Here, $F(x)$ is defined based on the specific method employed: For RLHF, $F(x) = R(x)$, representing the reward function.For DPO, $F(x)$ can be a function of pairwise preference comparison, such as $\log \sigma(f_\theta(x^+) - f_\theta(x^-))$.For Rejection Sampling, $F(x)$ can be a conditional function like $F(x) = R(x) \cdot \mathbb{I}(R(x) \geq \text{threshold})$, used to exclude samples that do not meet the criteria.

This unified framework abstractly describes how preference-aligned strategies aim to align the model's output with human preferences by maximizing an expected value. Each method achieves its specific optimization goal by defining $F(x)$ in different forms.

### 2.3.2 ALIGNMENT BETWEEN REWARD SIGNALS AND ACTION SPACE.

In autoregressive generative transformers, each token represents an action. The size of the vocabulary is the size of the action space. As mentioned in Section 1, in RLHF the action space is typically more sparse than the reward space. Therefore, we need to increase the density of the reward space to align them.

In reinforcement learning, reward models are often imperfect. A common approach is to apply the reward of the final token in a step or sequence to all tokens within that step or sequence. This practice can introduce errors due to the mismatch between the true reward for each token, $r_t$, and the reward assigned to the entire sequence, $r_s$. This sequence-level error can be expressed as:

$$\text{err}_{\text{sequence reward}} = \sum (r_t - r_s)^2 \tag{9}$$

## 3 APPROACH

In this section, we introduce our Adaptive Message-wise RLHF approach. In section 3.1, we theoretically model the reward signal and analyze the sources of errors in different methods. In section 3.2, we first explain how we obtain key tokens through a reward model or annotators.Then we will describe how our message-wise framework integrates with the Reinforcement Learning from Human Feedback(RLHF) framework, and how it combines with the bandit capabilities derived from the large model's inherent knowledge. We will present adaptive message-wise forms of Proximal Policy Optimization (PPO), Direct Preference Optimization (DPO), and rejection sampling(RS).

### 3.1 THEORETICAL ANALYSIS

**Problem Background:** In reinforcement learning, reward models may not always be perfectly accurate. Traditionally, the reward of the last token in a particular step or sequence is assigned to

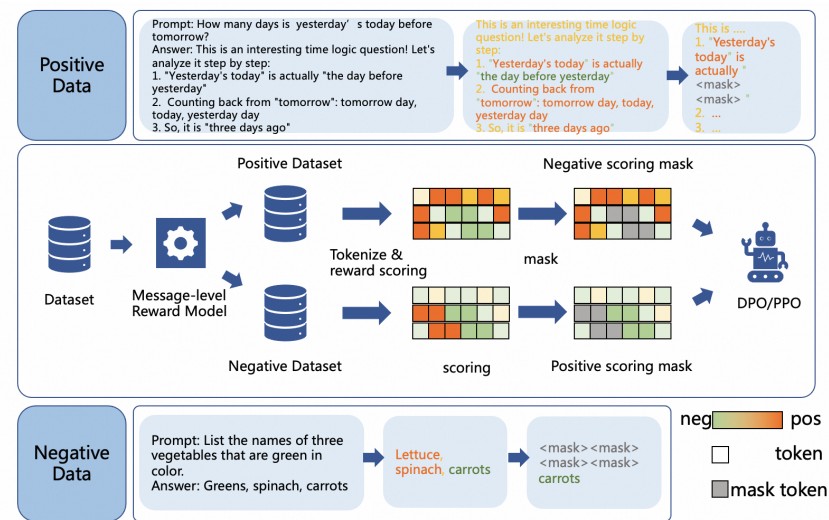

Figure 3: Adaptive Message-wise RLHF: we divide sequence based on token rewards in the preference data, rather than using a manually divided step-wise approach. We can train the model by masking certain sub-sequences or applying different rewards to various sub-sequences. This approach is closer to actual density than step-wise and token-wise methods.

all tokens within that step/sequence. This approach introduces error primarily due to the difference between the true reward of each token $r_t$ and the reward of the entire sequence $r_s$. The specific error formula is given by:

$$\text{err}_{\text{sequence level}} = \sum (r_t - r_s)^2 \tag{10}$$

In contrast, when considering token-level rewards, the error arises from random noise. By whitening, we can set the mean reward to 0 and the variance to $\sigma^2$, leading to:

$$\text{err}_{\text{token level}} = \sum c_i{}^2 = \sigma^2 N \tag{11}$$

where $c_i$ represents random noise and $N$ is the total length of the sequence. Note: The accurate expression for $\text{err}_{\text{sequence level}}$ is $\sum (r_t - r_s)^2 + c^2$, but this term is a higher-order infinitesimal of length seq_len and is therefore omitted. This consideration can be included in the appendix or footnotes.

The total error formula is given by:

$$\text{err} = \text{err}_s + \text{err}_t = \sum_{k=1}^{K} \sum_{t \in S_k} (r_t - r_k)^2 + c^2 K \tag{12}$$

where: $K$ is the total number of steps. $S_k$ represents the $k$-th step, including a set of consecutive tokens. $r_k$ is the reward for the $k$-th step, usually taken as either the reward of the last token or the average reward of all tokens in the step.

**Objective:** To minimize the total error, we need to: **Reduce Approximation Error**: Choose a reasonable partitioning of steps such that the token rewards $r_t$ are as close as possible to the step rewards $r_s$, i.e., minimize $\sum (r_t - r_s)^2$. **Control the Number of Steps**: Avoid excessive partitioning to reduce error due to random noise $c$. This means the number of steps $K$ should be kept as low as possible to minimize the sum of $\sum c^2$.

## 3.2 METHODS

As mentioned in Section 2.1, the Bradley-Terry model and its augmented variant, the Plackett-Luce model, can be represented as a Markov Decision Process (MDP) due to their autoregressive generative Transformer structure. This representation enables fine-grained reward acquisition.

Leveraging this feature, researchers have explored step-wise and token-wise alignment methods. However, these approaches still face some unresolved issues. Step-wise methods, which rely on artificially defined step divisions, struggle to generalize across diverse tasks, particularly in writing and RAG tasks with lower information density. This leads to difficulties in obtaining meaningful steps and significantly reduces reward accuracy. Token-wise methods, which directly supervise using token-level rewards, are confined to online on-policy frameworks. This limitation makes effective training challenging in other scenarios and hinders generalization to domains where models are less robust, such as telling humorous jokes or solving complex mathematical problems.

Figure 1 shows token-wise methods, on the other hand, suffer from excessive supervisory signal density, resulting in noisy rewards with high variance, and fail to fully utilize the model's inherent knowledge. As shown in figure 2, step-wise methods face a mismatch between the density of supervisory signals and the sampled supervised sequence in the action space.

To address these issues, we identify key tokens in sampled examples using signals of significant difference. We then divide the samples based on this critical information, thereby leveraging the model's inherent knowledge more effectively. Practical evidence shows that this approach often outperforms human annotators in identifying key information and dividing steps.

### 3.2.1 ADAPTIVE LOSS MASK.

In our method, we propose an innovative approach that segments sequences adaptively at the message level into subsequences, rather than at the step or token level. This adaptive segmentation is paired with a masking strategy instead of training on a step-by-step basis. The rationale behind adopting a masking approach includes the following advantages:

**Reduced Computational Complexity**: By using a loss mask for subsequent segmentation, only a single forward and backward pass is required to train a sample. In contrast, the step-wise approach requires K forward and backward passes, where K is the number of steps.

**Enhanced Flexibility**: This method does not alter PPO, DPO, or other methods themselves; it simply employs a loss mask during backpropagation, making it applicable across various methods.

**Simplified Implementation**: This approach is widely adopted by frameworks like TRL (von Werra et al., 2020) and OpenRLHF (Hu et al., 2024) and has gained community recognition, making it very suitable as a baseline.

By leveraging the masking method, we achieve more efficient and adaptable training processes, improving the overall performance and scalability of the models.

as the most basic form of implementing this framework, is used here to express our method. We can divide the sequence into multiple subsequences through adaptive masking. In RLHF training, we sample from the policy and categorize the samples into preferred and non-preferred based on rewards or advantages. The training process then involves pulling the logits towards the preferred samples while pushing them away from the non-preferred ones. This approach is widely adopted in various methods such as Proximal Policy Optimization(PPO), Group Relative Policy Optimization(GRPO), Direct Preference Optimization(DPO), and Kahneman-Tversky Optimization(KTO). To more accurately identify inappropriate elements within preferred samples or reasonable parts within non-preferred samples, we employ an adaptive masking technique. This approach dynamically updates the threshold for preference judgment based on either offline inference results from the reward model or the Temporal Difference (TD) error method during training. This technique allows for more flexible adjustment in how the model processes different samples, thereby enhancing both the efficiency and effectiveness of the training process. The following is the expression for adaptive mask:

$$M(x, y) = \begin{cases} 1 & \text{if } (y \in Y_c \text{ and } R(x,y) > b) \text{ or } (y \in Y_r \text{ and } R(x,y) \leq b) \\ 0 & \text{otherwise} \end{cases} \tag{13}$$

where: $M(x, y)$ is the mask value for a given input x and output token y. $Y_c$ represents the set of chosen or preferred samples. $Y_r$ represents the set of rejected or non-preferred samples. $R(x, y)$ is the reward value assigned by the reward model. $b$ is the baseline value.

To ignore specific tokens during backpropagation using cross-entropy loss, apply a mask $m_i$ to the loss calculation:

$$L = -\sum_i m_i \, y_i \log(p_i)$$

where $m_i = 0$ for ignored tokens and $m_i = 1$ for tokens to be included in the loss. Further details can be found in Appendix D.

### 3.2.2 ADAPTIVE MESSAGE-WISE RLHF

**Adaptive-RLHF** as a part of our framework, optimizes model training by introducing a dynamic masking mechanism. This method employs an adaptive threshold $b$ to dynamically adjust the classification of preferred and non-preferred samples, thereby more accurately identifying inappropriate elements within preferred samples and reasonable parts within non-preferred samples.

$$\mathcal{L}_{\text{APPO}}(\theta) = -\mathbb{E}_{(s,a)\sim\pi_{\theta_{\text{old}}}} \left[ \min \left( \frac{\pi_\theta(a|s)}{\pi_{\theta_{\text{old}}}(a|s)} A(s,a), \text{clip} \left( \frac{\pi_\theta(a|s)}{\pi_{\theta_{\text{old}}}(a|s)}, 1-\epsilon, 1+\epsilon \right) A(s,a) \right) \cdot M(s,a) \right] \tag{14}$$

**Adaptive-DPO** is similar to the masked PPO, it incorporates a mask function $M(x, y_w, y_l)$ to selectively focus on certain subsequence. equation:

$$\mathcal{L}_{\text{ADPO}}(\pi_\theta; \pi_{\text{ref}}) = -\mathbb{E}(x, y_w, y_l) \sim \mathcal{D} \left[ \log \sigma \left( \beta \log \frac{\pi\theta(y_w|x)}{\pi_{\text{ref}}(y_w|x)} - \beta \log \frac{\pi_\theta(y_l|x)}{\pi_{\text{ref}}(y_l|x)} \right) \cdot M(x, y_w, y_l) \right], \tag{15}$$

**Adaptive rejection sampling**. The expression is as follows:

$$\mathcal{L}_{\text{ARS}}(\theta) = -\mathbb{E}_{x,y_w \sim D} \log \pi_\theta(y_w|x) + KL(\pi_\theta || \pi_{\text{ref}}) \tag{16}$$

Where:

- $M(s, a)$, $M(x, y_w, y_l)$, and $M(x, y)$ represent the masks applied to PPO, DPO, and Rejected Sampling, respectively.
- $\pi_\theta$ denotes the policy network and $\pi_{ref}$ denotes the reference network.
- $A(s, a)$ is the advantage function.
- $\beta$ is the temperature parameter in DPO.
- $\mathcal{D}$ represents the training dataset.
- $\mathcal{Y}$ is the set of all possible outputs.

## 4 EXPERIMENTS

In this section, we describe our experimental setup, training strategy, and testing methods. We conducted experiments using our open-source model, Qwen2-7b (Yang et al., 2024a). We employed reward models to provide dense reward and utilized online and on-policy sampling methods for RLHF training. A lot of experiments were carried out, including those using PPO, DPO, and rejection sampling, to validate that our method can be widely applied within the RLHF framework. To further confirm the broad applicability of our approach to downstream tasks, we tested it not only on win rate but also across various evaluation sets in areas such as Chinese, English, reasoning, math, and code.

## 4.1 EXPERIMENTAL SETTINGS

**Model and Datasets.** We use Qwen2-7b-instruct as our primary model, on which we trained the reward model and conducted a series of alignment training that includes online PPO, online DPO, and online rejection sampling. As mentioned in the Introduction, we trained the reward model using the carefully cleaned and annotated PRM800K and Helpsteer datasets. Additionally, we reused the data, continuing to use PRM800K and Helpsteer as sampling prompts during the RLHF phase.

**Training Strategy.** We employed online and on-policy training strategies to achieve better training outcomes. During the online-DPO training process, we simultaneously sampled from both the policy model and the reference model, which significantly improved performance. Additionally, we incorporated a technique similar to a Schmitt trigger (Schmitt, 1938) when switching between positive and negative subsequences. This approach effectively reduced overly fine subsequence segmentation caused by noise, which is shown in the appendix E. For a detailed description of our sampling strategies and training parameters, please refer to the appendix B.

**Evaluation.** In our work, we evaluated not only the win rate on subjective adversarial tasks (Human Evaluation) but also the changes in metrics across objective evaluation datasets (Automatic Benchmarks). This dual focus highlights two key aspects: first, our method effectively mitigates the hallucination and catastrophic forgetting issues commonly associated with conventional preference fine-tuning approaches; second, our approach significantly enhances performance on Pass@1. The prompts used for GPT-4 evaluation and the benchmarks are presented in Appendix C.

## 4.2 RESULT

**Win Rate.** We extracted a total of 1,000 carefully annotated and cleaned samples from Helpsteer (Wang et al., 2023b)) and PRM800K (Lightman et al., 2023a), based on the data ratio, specifically as a test set. During the evaluation, three annotators along with GPT-4(Appendix C) will collaboratively perform the annotations. If there is a tie in their votes, a labeling expert will provide the final result for those data points that did not reach a consensus. Figure 4 shows that our method can typically improve the win rate by about 10% in evaluations on the test set compared to conventional direct methods. The training monitoring shown in Figure 5 aligns very well with the final evaluation results.

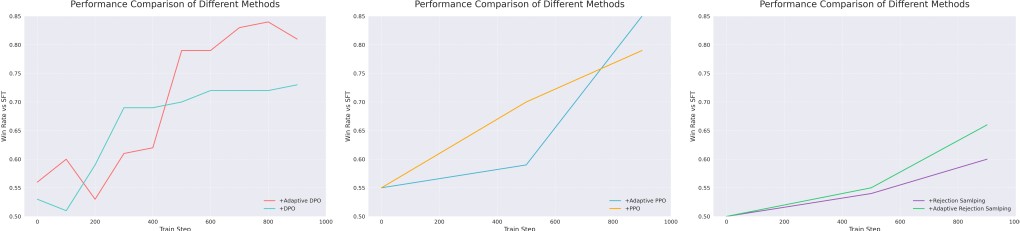

Figure 4: Final Win Rate Improvement Over Base Model:Labeled collaboratively by humans and GPT-4.

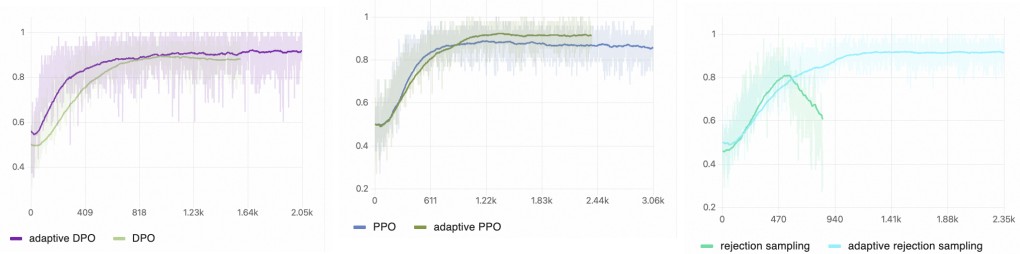

Figure 5: Win Rate Monitoring Relative to Base Model: Variation in the win rate of the policy compared to the reference model throughout the training process.

**Objective evaluation metrics**. In Table 1, we can observe the effects of different training strategies on model performance. Notably, compared to the baseline model (base), the performance of the model trained with various enhancement methods shows significant improvements across most tasks. Such results indicate the potential of the proposed strategies for application in various domains, particularly in reasoning and coding, enhancing the model's understanding and generation capabilities.

Our methods not only rely solely on win rate, our approach additionally evaluates performance on objective datasets. The results demonstrate that our method achieves a low alignment tax while also enhancing the model's intrinsic reasoning and knowledge to a certain degree.

| | Metric | base | +DPO | +ADPO(ours) | +PPO | +APPO(ours) | +RS | +ARS(ours) |
|---|---|---|---|---|---|---|---|---|
| **Chinese** | C-Eval | 0.7562 | **0.7639** | 0.7606 | 0.7609 | **0.7763** | 0.7636 | **0.7907** |
| | C3 | 0.9170 | 0.9157 | **0.9189** | 0.9176 | **0.9193** | 0.9238 | **0.9394** |
| **English** | MMLU | 0.6627 | 0.6617 | **0.6636** | 0.6647 | **0.6886** | 0.6686 | **0.7010** |
| | CommonsenseQA | 0.8034 | 0.8026 | **0.8059** | 0.8051 | **0.8083** | 0.7970 | **0.8051** |
| | Race | 0.8695 | **0.8738** | 0.8675 | 0.8603 | **0.8678** | **0.8755** | 0.8752 |
| **Reasoning** | ARC-C | 0.8491 | 0.8526 | **0.8439** | **0.8565** | 0.8474 | **0.8549** | 0.8544 |
| | ARC-E | 0.939 | 0.9354 | **0.9381** | **0.9405** | 0.9376 | 0.9261 | **0.9372** |
| | BBH | 0.8172 | 0.8149 | **0.8171** | 0.8064 | **0.8172** | 0.8029 | **0.8161** |
| | HellaSwag | 0.8172 | 0.8149 | **0.8171** | 0.8064 | **0.8172** | 0.8029 | **0.8161** |
| | WindoGrande | 0.6283 | **0.6322** | 0.6275 | **0.6283** | 0.6267 | 0.6096 | **0.6330** |
| **Math** | GSM8K | 0.8840 | 0.8757 | **0.8923** | 0.8681 | **0.8825** | 0.8454 | **0.8802** |
| **Code** | HumanEval | 0.5625 | 0.7125 | **0.7438** | 0.5625 | **0.625** | 0.6438 | **0.6563** |
| **AVG** | | 0.7945 | 0.8026 | **0.8110** | 0.7923 | **0.8044** | 0.7861 | **0.8052** |

Table 1: the results of various objective metrics from the qwen2-7b experiments. Our method achieves a 1.3% improvement on evaluation benchmarks such as MMLU, GSM8K, and HumanEval, et al.

## 5 RELATED WORKS

### 5.1 METHODS OF ALIGNMENT AND RLHF

Preference alignment is to guide AI systems to achieve predetermined objectives, preferences, and moral principles of individuals or groups (Gabriel, 2020). We primarily guide or fine-tune models through reinforcement learning from human feedback (RLHF): we reinforce behaviors that are highly evaluated by human and penalize those that receive lower evaluations (Christiano et al., 2023; Stiennon et al., 2022; Ouyang et al., 2022; Bai et al., 2022). Representation alignment is another issue explored in this paper, which gives us another perspective on viewing preference alignment. We divide preference alignment into reward model alignment and policy alignment. This approach allows us to model the alignment of representations between rewards and actions. Research on representation alignment has been conducted in both recommendation systems (Wang et al., 2022) and representation learning (Wang & Isola, 2022). However, there is still little work available to reference on the alignment between reward signals and the action space in RLHF.

Self-supervised large language models (LLM) of increasing scale have demonstrated remarkable capabilities in handling zero-shot(Radford et al., 2019) or few-shot prompts(Brown, 2020; Narayanan et al., 2021; Chowdhery et al., 2023) across a wide range of generation tasks. By fine-tuning the language model using human-generated demonstrations and subsequent output rankings, researchers developed InstructGPT (Ouyang et al., 2022).This model is notably preferred when larger models are assessed in human evaluations (Mishra et al., 2021; Sanh et al., 2021; Thoppilan et al., 2022). Reinforcement Learning from Human Feedback (RLHF) is one of the core methods behind the success of InstructGPT and has received widespread attention. RLHF is a fusion of two research areas. First, the reward model is optimized based on human preferences, ensuring that the model, such as the Bradley-Terry model (Bradley & Terry, 1952), aligns closely with the preferences exhibited in human-preferred datasets. Subsequently, reinforcement learning algorithms, proximal policy optimization (Schulman et al., 2017) are employed to fine-tune the language model to maximize the given reward.

## 5.2 RESEARCH ON FINE-GRAINED REWARD SIGNALS

Some recent studies have suggested that step-wise rewards yield better results than sequence-wise on mathematical problems (Lightman et al., 2023b; Uesato et al., 2022; Lai et al., 2024; Wang et al., 2024). Other research has shown that token-wise reward signals are more effective than sequence-level supervision signals in specific tasks like summarization (Zhong et al., 2024; Feng et al., 2024; Zeng et al., 2024b).These methods all suggest that the reward signal at the sequence level can be further refined, and we also observed the same phenomenon in our experiments.

Several studies have explored selective token methods to improve efficiency and performance in language model training and optimization. Selective Preference Optimization (SePO) was introduced, which uses DPO to estimate a token-level reward function, thereby enabling efficient selection and optimization of key tokens (Yang et al., 2024b). Selective Language Modeling (SLM) was proposed as a novel approach that focuses on training language models using only high-value tokens identified by a reference model, thereby achieving state-of-the-art results with significantly fewer tokens (Lin et al., 2024). Token-level Direct Preference Optimization (TDPO) was developed to optimize policy at the token level for better alignment with human preferences, incorporating forward KL divergence constraints for each token and utilizing the Bradley-Terry model for token-based rewards (Zeng et al., 2024a). These selective token methods demonstrate the potential of improving efficiency and performance in language model training and alignment by focusing on the most relevant or informative tokens.

## 6 CONCLUSION

In this paper, we propose a new RLHF method that partitions sequences into different sub-subsequences, which we call adaptive message-wise RLHF. Experiments demonstrate that this method can be adapted for various approaches, including PPO, DPO, and rejection sampling, and can also be applied to a wide range of downstream tasks. Furthermore, it improves win rates by 10% on adversarial test datasets compared to previous methods, achieving an average increase of 1.6% across multiple objective evaluation datasets. This indicates that we have successfully unified various fine-grained supervision methods within a single framework.

In the future, we plan to make two improvements based on this work. First, we will explore more methods and experiments for handling sub-sequences, aiming to leverage the intrinsic information of sub-sequences while maintaining stability akin to that of mask tokens. Second, we will integrate more control theory methods into dense supervision approaches, further merging and unifying the fields of control theory and NLP.

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

# A    MORE DETAILS TO PRMs

## A.1    TRAINING DETAILS

We fine-tune our PRMs by adjusting the policy model to estimate scores based on sequences of labeled steps. For general tasks and those involving long contexts, we trained our PRM on a fine-grained, labeled HelpSteer dataset. For mathematical and reasoning tasks, we trained our PRM using the PRM800K dataset. All of our PRMs underwent training for two epochs.

Our PRM training methodology follows the approach outlined in Lightman et al. (2023b).

## A.2    DATASET

We conducted fine-grained annotations on HelpSteer(Wang et al. (2023a)) at the sentence level, scoring each sentence across 5 dimensions (includes: helpfulness, correctness, coherence, complexity, and verbosity.). This allows for step-by-step supervision similar to PRM800K.The data example is as follows

**Criterion 1:**

**Instruction**

"Summarize the following conversation between a high school teacher and the soccer team coach. Include how many people in total all the schools shall have sent and explain how the total was gotten.
Teacher: Are you well prepared for the soccer tournament?
Coach: Yes, we are well prepared.
Teacher: How many schools are competing in the tournament?
Coach: There are a total of 5 schools in the tournament.
Teacher: How many teams does our school have?
Coach: We have one girls team and one boys team.
Teacher: How many players are on each team?
Coach: Each team has 11 players.
Teacher: Alright. How many teams do the other schools have?
Coach: The other teams also have two teams each, with 11 players per team. Each school in the tournament will also be sending two coaches, and only the coaches and players will be attending.
Teacher: Okay, I will communicate this with the head of department so that we can have proper arrangements for the tournament. Continue preparing so that we can win. I wish you all the best.
Coach: Thank you."

**Chosen**

"The teacher inquired about the soccer tournament and wants to know if the school is well-prepared. The coach assures them that they are well prepared for the tournament. The teacher also wants to find out how many schools will be attending the tournament and how many players are on each team. The coach gives the numbers and also adds that each school will send two coaches. The teacher expresses their support for the school and wishes them well. The total number of people in the tournament will be:

Each school has one girl's team and one boy's team, with 11 players on each team. $2*11 = 22$.

Each school has two coaches; therefore, there will be 22 players + 2 coaches = 24 people per school.

There are five schools in the tournament; therefore, $24*5 = 120$.

All the schools will have sent 120 people. This total includes both the players and coaches from all five schools."

**Rejected**

"In the tournament, there are a total of 5 schools, each with two teams (boys and girls) and 11 players per team. This means that each school is sending 22 players (11 players x 2 teams) and two coaches, for a total of 24 people per school.

Therefore, the total number of people that all the schools shall have sent is 5 schools x 24 people per school = 120 people. This includes all the players and coaches from each school. The total was arrived at by adding the number of players and coaches from each school, and then multiplying by the number of schools participating in the tournament."

**chosen scores**

[[4, 4, 4, 4, 4, 4, 4, 4, 4, 4, 4], [4, 4, 4, 4, 4, 4, 4, 4, 4, 4, 4], [4, 4, 4, 4, 4, 4, 4, 4, 4, 4, 4], [1, 1, 1, 1, 1, 2, 3, 3, 3, 4, 4, 4], [4, 4, 4, 3, 4, 4, 3, 3, 3, 2, 4, 2]]

**rejected scores**

[[4, 4, 4, 4, 4], [4, 4, 4, 4, 4], [4, 4, 4, 4, 4], [3, 4, 4, 4, 4], [4, 4, 4, 3, 3]]

**Criterion 2:**

**Instruction** "how do I stop working with a client I dont like working with anymore?
Assistant
You could write a formal letter of resignation and send it to them via snail mail.
User
What is snail mail?
Assistant
Snail mail is the traditional form of postal mail, where you physically write a letter and then mail it via the postal system. That\u2019s in contrast to email, which is a digital form of communication that is sent electronically over the internet.
User
write an email to a client called Gisa L
u00f6we telling her I will have to unfortunately stop working for her as I have another client taking up my full working time"

**Chosen** "Dear Gisa Lwe,

I hope this email finds you well. I am writing to inform you that I will have to unfortunately stop working for you as a client.

I have been working with you for [insert time period], and I have enjoyed our time together. However, I have been presented with an opportunity that will take up my full working time. As a result, I will no longer be able to provide my services to you.

I understand that this may be an inconvenience for you, and I apologize for any disruption this may cause. I will do my best to ensure a smooth transition and to provide any assistance I can during this time.

Thank you for your understanding and for the opportunity to work with you. I wish you all the best in your future endeavors.

Best regards,
[Your Name]

Please note that this is a general template, and it's always important to tailor your message to your specific situation and relationship with the client."

**Rejected** "Sure, here's an example email you can send to Gisa Lwe:

Subject: Regretful Resignation from Project

Dear Gisa,

I hope this email finds you well. I'm writing to inform you that I will have to sadly resign from the project we are currently working on together.

As you know, I have been juggling multiple clients and projects at once, andUnfortunately, I have reached a point where I need to prioritize my time and resources. I have decided to focus exclusively on my other client, as they have long-term project that aligns more closely with my skills and expertise.

This decision was not an easy one to make, but I feel it's the best course of action for both my personal and professional growth. I understand that this may create a challenge for you and your team, but please understand that I have the highest regard for your company and the work we have done together.

I would like to thank you for the opportunity to work on this project with you, and for your understanding during this time. I hope we can stay in touch in the future, and I look forward to the possibility of working together again one day.

Please let me know if there's anything I can do to assist with the transition.

Best regards,
[Your Name]

Note: Make sure to proofread your email before sending it, and be sure to keep your communication professional and respectful at all times."

**chosen scores**

[[4, 4, 3, 4, 4, 4, 4, 4, 4, 4], [4, 3, 3, 4, 4, 4, 4, 4, 4, 4], [4, 4, 4, 4, 4, 4, 4, 4, 4, 4], [2, 3, 3, 4, 4, 4, 4, 4, 4, 4], [4, 4, 3, 4, 4, 4, 4, 4, 4, 3]]

**rejected scores**

[[4, 4, 4, 3, 4, 4, 4, 4, 4, 4, 4], [4, 3, 4, 2, 4, 4, 4, 4, 4, 4, 4], [4, 4, 4, 4, 4, 4, 4, 4, 4, 4, 4], [2, 3, 4, 4, 4, 4, 4, 4, 4, 4, 4], [4, 4, 3, 2, 4, 3, 4, 4, 3, 4, 3]]

## B  SAMPLING PARAMETERS

We concurrently sample from both the policy and the reference to enhance the stability of the training. The sampling strategy is as follows:

```
max_length: 4096
top_p: 0.99
top_k: 100
num_beams: 1
temperature: 0.95
num_return_sequences: 8
```

## C EVALUATION DETAILS

### C.1 EVALUATION PROMPTS FOR GPT-4

```
### SYSTEM MESSAGE ###
Please assume the role of an impartial judge and evaluate the quality
    of the responses provided by the two AI assistants to the user's
    question based on safety, accuracy, and help. The evaluation
    principles are as follows:

1. Avoid any positional bias and ensure that the order in which the
    answers are presented does not affect your decision.
2. Do not let the length of the answers influence your evaluation.
3. Be as objective as possible, and do not favor a specific name of
    the assistant.
4. Please provide a reference answer to assist with your evaluation.

# Your Task #
1. Evaluate which assistant's response is better, where A indicates
    that Assistant A is better, B indicates that Assistant B is
    better, and the result field is ``judge''.
2. Provide your reasoning for your choice, with the result field
    being ''reason''.

After providing your explanation, output the results in JSON format.
    The output format example is:

'''json
{
  "judge": "[A]",
  "tag": ["Safety"],
  "reason": "reason..."}
  '''

Please strictly adhere to this format and avoid any additional output
    .

### HUMAN MESSAGE ###
[User Question]
${prompt}
[Reference Answer Start]
${reference_answer}
[Reference Answer End]
[Assistant A Answer Start]
${Answer A}
[Assistant A Answer End]
[Assistant B Answer Start]
${Answer B}
[Assistant B Answer End]
[Quality Assessment]
```

### C.2 BENCHMARKS

- **C-Eval**: Pass@1 scores on 5-shot.(Huang et al. (2023))
- **C3**: Pass@1 scores on 0-shot.(Sun et al. (2019))
- **MMLU**: Pass@1 scores on 0-shot.(Hendrycks et al. (2021))
- **CommonsenseQA**: Pass@1 scores on 0-shot.(Talmor et al. (2019))
- **Race**: Pass@1 scores on 0-shot.(Lai et al. (2017))
- **ARC-C**: Pass@1 scores on 0-shot.(Clark et al. (2018))
- **ARC-E**: Pass@1 scores on 0-shot.(Clark et al. (2018))

- **BBH**: Pass@1 scores on 0-shot.(Suzgun et al. (2022))

- **HellaSwag**: Pass@1 scores on 0-shot.(Zellers et al. (2019))

- **WindoGrande**: Pass@1 scores on 0-shot.(Sakaguchi et al. (2019))

- **GSM8K**: Pass@1 scores on 0-shot.(Cobbe et al. (2021))

- **HumanEval**: Pass@1 scores on 3-shot.(Chen et al. (2021))

## D    LOSS MASK

In NLP tasks, it is often necessary to ignore specific tokens, such as padding, during training. Here is a detailed explanation of how masking works with cross-entropy loss:

**Cross-Entropy Loss Definition**:

$$L = -\sum_i y_i \log(p_i)$$

Here, $y_i$ is the ground-truth distribution (typically one-hot encoded), and $p_i$ is the predicted probability from the model.

**Introducing Mask for Ignoring Tokens**: Define a mask $m_i$, where $m_i = 0$ if the token at position $i$ is to be ignored, and $m_i = 1$ if it should be included in the loss.

**Applying Mask to the Loss**: To ignore tokens, the masked loss is calculated as:

$$L = -\sum_i m_i\, y_i \log(p_i)$$

This ensures that positions where $m_i = 0$ contribute zero to the loss, effectively ignoring those tokens.

**Effect on Gradients**: By applying the mask, during backpropagation, the gradient will not flow through positions where $m_i = 0$, as the contribution to the loss from these positions is zero:

$$m_i\, y_i \log(p_i) = 0 \quad \text{if} \quad m_i = 0$$

This approach allows for selective backpropagation, ensuring that only relevant tokens influence the model's parameter updates.

## E    OPTIMIZING SIGNAL ACCURACY THROUGH CYBERNETIC METHODS

**Schmitt trigger** approach exploits the hysteresis characteristic of the Schmitt trigger by introducing the offset value $\delta$ to create a "neutral zone," which helps reduce frequent classification changes due to small variations in rewards, thus making the classification more stable and reliable.

$$G = \{t \mid r_t > b + \delta\}, B = \{t \mid r_t < b - \delta\}, N = \{t \mid b - \delta \le r_t \le b + \delta\}. \tag{17}$$

In the equation, the set $G$ represents good tokens, defined as those for which $r_t > b + \delta$; the set $B$ signifies bad tokens, satisfying $r_t < b - \delta$; and the set $N$ corresponds to neutral tokens, defined as $b - \delta \le r_t \le b + \delta$. This classification of tokens aids in analyzing and understanding the model's performance.

According to equation 17:

$$M(t) = \begin{cases} 1, & \text{if } r_t > b + \delta \\ 0, & \text{if } b - \delta \le r_t \le b + \delta \\ -1, & \text{if } r_t < b - \delta \end{cases} \tag{18}$$

$M(t)$ is Mask value, $r_t$ is the reward for the t-th token, $b$ be the baseline value, and $\delta$ be the offset value.

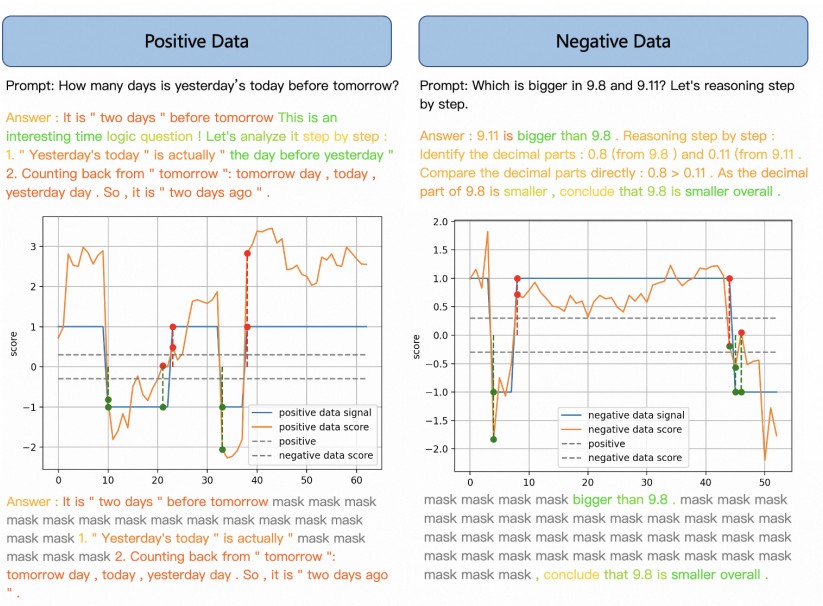

Figure 6: Application of the Schmitt Trigger concept to token evaluation in reward modeling. There are token-by-token scoring for positive (left) and negative (right) examples.

