# OpenReview forum: "Adaptive dense reward:Understanding the Gap Between Action and Reward Space in Alignment"
_ICLR.cc/2025/Conference — ICLR 2025 Conference Withdrawn Submission_

### Official Review · Reviewer_7YEj · 2024-10-27

**Soundness:** 2
**Presentation:** 1
**Contribution:** 2
**Rating:** 3
**Confidence:** 4

**Summary:**

The paper introduces Adaptive Message-wise RLHF, a method enhancing reinforcement learning in language models by dynamically adjusting reward signals at the subsequence level.
The method is introduced to improve alignment accuracy and mitigate issues like hallucinations and forgetting across diverse tasks.
Experiments are conducted on multiple datasets.

**Strengths:**

The method is evaluated on a variety of benchmarks.

**Weaknesses:**

in lines 272 - 274, the authors claim "Bradley terry model can be represented as an MDP", this is incorrect.

the paper needs spellcheck for e.g., typos or missing spaces. For a few examples: line 76, 292, 298, equation 13.

missing references when citing existing papers. The authors should not assume their readers to find references to the terms/concepts needed in their paper.

Several papers have been cited multiple times: step-dpo, token-level dpo, Uesato et al. , Lightman et al.,  and helpsteer.

Figure 2-4 are not referred to in the main text. I failed to understand the messages of those figures.

Some notations are redundant: e.g., y_w, y_l, y_c, y_r

The paper's quality can be largely improved by improving the presentation.

Citing HER in the section of related work is a bit wild. There are many other related works to cite in the field of better credit assignments in RL.

**Questions:**

Why are some experiments truncated? Can the authors explain their efforts that could guarantee the comparisons are fair to all algorithms?

How many repeated runs do the authors conduct in concluding the results? Are the improvements statistically significant?

How do the authors determine the values of delta?

Is the b learnable? How do the authors implement this learnable parameter to stabilize the learning process? --- Learning with a learnable parameter sounds like shooting a moving target, and would be challenging intuitively.

Equation 11 and equation 16 present different possible values for M.

In Table 1, could the authors also compare with other methods like step-dpo, token-dpo?

---

> ### Author Response · Authors · 2024-12-02
> **Response to Reviewer 7YEj**
>
> We would like to sincerely thank Reviewer 7YEj for providing a detailed review with insightful suggestions.
>
> > **W1**: in lines 272 - 274, the authors claim "Bradley terry model can be represented as an MDP", this is incorrect.
>
> Thank you for pointing that out. The intent here is to convey that autoregressive generative models can be viewed as MDPs, which is widely considered the foundation for applying RL to LLMs. We have revised the wording to: ```can be represented as a Markov Decision Process (MDP) due to their autoregressive generative Transformer structure.```
>
> > **W2-8**:
>
> We truly apologize for writing quality.
> * We have rechecked the formatting of the entire paper. Improvements include a comprehensive check and correction of spacing, formulas, and citations.
> * The introduction now contains the research objectives, significance, and main contributions of this work. Parts that duplicated the literature review have been removed or relocated to Section 5.
> * The literature review has been significantly condensed, focusing on a comparison of the most relevant work. Section 5.1 now covers alignment（As reviewer GTLV's opinion）and RLHF, while Section 5.2 discusses research on fine-grained reward signals.
>
> > **Q1-2**: Why are some experiments truncated? Can the authors explain their efforts that could guarantee the comparisons are fair to all algorithms?
>
> Some experiments were stopped early when both algorithms reached a performance plateau, indicating further training offered minimal improvement. This was done to conserve computational resources.
>
> To ensure fairness in comparisons, we took several measures:
>
> * Consistent Data and Code: All algorithms were trained and evaluated on the same dataset using the same codebase.
> * Controlled Random Seed: A fixed random seed (42) was used across all experiments to minimize the impact of randomness and ensure reproducibility.
> * Multiple Runs: Each algorithm was run twice, with results consistent within 0.1%, except for minor variations due to multi-machine communication and flash-attention. This consistency reinforces the reliability of the reported results and minimizes the potential bias introduced by early stopping. We are happy to provide the results from all runs if requested.
>
> > **Q3-4**: How do the authors determine the values of delta? Is the b learnable?
>
> Following the online-DPO training procedure, δ was set to 0.1, consistent with the threshold.
>
> We have two methods for setting the baseline(b).
>
> 1. Offline sampling to calculate the mean reward score, and then using 0 as the baseline after subtracting this mean from the reward model's scores.
>
> 2. The second method starts by using 0 as the baseline to distinguish positive and negative rewards at step 0 of the experiment. Simultaneously, we initialize an array to store the current step and the mean reward. As training progresses, we dynamically update this mean based on the reward scores given by the reward model, and use this updated mean as the baseline.
>
> In our experiments, we found the first method to be more effective. All experiments in this paper utilize the first method.
>
> > **Q5**: Equation 11 and equation 16 present different possible values for M.
>
> The meanings of 'M' in Equation 11 and Equation 16 are distinct. In Equation 11, 'M' represents a loss mask, taking binary values of 0 and 1. Conversely, 'M' in Equation 16 signifies an evaluation of corpus quality, categorized as positive, negative, or neutral.
>
> > **Q6**: In Table 1, could the authors also compare with other methods like step-dpo, token-dpo?
>
> I've attempted a comparison, but I haven't been able to find a fair comparison method. Both methods' authors only provide baselines on specific tasks (like math and summarization). Step-DPO's performance is significantly affected by the choice of step divisions, and token-DPO fails to converge on general tasks. Comparing them may not be essential to the core contribution of this paper.

---

### Official Review · Reviewer_BZ8y · 2024-10-28

**Soundness:** 2
**Presentation:** 1
**Contribution:** 1
**Rating:** 3
**Confidence:** 4

**Summary:**

This paper addresses the RLHF (Reinforcement Learning with Human Feedback) task in Large Language Models (LLMs). The authors identify the limitations of the reward model, particularly its inaccuracies when responses contain erroneous tokens. They introduce a method called adaptive message-wise RLHF, which detects essential information and employs a mask matrix to enhance training. Experimental results in certain scenarios demonstrate the proposed method's effectiveness.

**Strengths:**

1.	The authors present an approach to overcoming limitations in RLHF.

2.	The method demonstrates efficiency in specific scenarios.

**Weaknesses:**

1.	Limited Contribution: The main contribution, calculating the mask value for each instance in Eq. 11, seems insufficiently significant.

2.	Questionable Effectiveness:

    a.	The preference dataset might contain biased data, potentially causing spurious correlations between the reward and outcome, which the proposed method might exacerbate.

    b.	Eq. 11 may not accurately identify the essential parts of the response, possibly disrupting training.

    c.	Eqs. 12 and 13 might overlook parts of the responses, leading to a loss of fluency and coherence.

3.	Poor Writing Quality: The paper's writing is subpar. For instance, Section 2 includes excessive, unnecessary related work.

4.	Inconsistent Notations: For instance, $y$ represents the whole response in Eq. 6 but seems to denote a token in Eq. 11. Similar confusion exists between $y_w$ in Eq. 6 and $y_{win}$ in Eq. 8.

5.	Unimpressive Experimental Results: The results lack significance. Only Qwen2-7b-instruct was used for evaluation, and more models need testing.

6.	Typos and Errors: There are typographical errors, such as $\pi\theta$

**Questions:**

1.	What does the title imply? Could you explain the concept of the gap between action and reward space alignment?

2.	How is the threshold b in Eq. 11 determined? Since it's adaptive, how can it be optimized in real-world applications?

3.	Have you experimented with alternative model architectures?

4.	What are the additional computational costs associated with this method?

---

> ### Author Response · Authors · 2024-12-02
> **Response to Reviewer BZ8y [Part 1]**
>
> We are grateful to the reviewer BZ8y for careful reading of our manuscript and valuable feedback.
> > **W1**: Limited Contribution: The main contribution, calculating the mask value for each instance in Eq. 11, seems insufficiently significant.
>
> That's an excellent suggestion. We've therefore expanded the presentation of our theoretical analysis.
>
> * The introduction now contains the research objectives, significance, and main contributions of this work. Parts that duplicated the literature review have been removed or relocated to Section 5.
> * To provide a clearer problem definition, we've added Section 2.3. In Section 2.3, we provide a detailed definition of the problem. In Section 2.3, we define and differentiate between the two types of alignment discussed in this paper. We clarify that the alignment referred to in the title is preference alignment (Section 2.3.1) and also explain representational alignment (Section 2.3.2, noting the need for a denser reward signal to align the action space).
> * In Section 3.1, we identify potential sources of error when the reward signal is sparse, clarifying why our approach of identifying key information within the sequence can effectively divide subsequence and reduce errors.
>
>
> In Section 3.1, we define the supervisory signal error. Traditional methods assign the reward of a step to all tokens within that step/sequence. The error in this method stems from the discrepancy between the overall reward for the sequence $r_s$ and the reward assigned to individual actions/tokens $r_t$. The error is formulated as:$$\text{err}_{\text{sequence level}} = \sum (r_t - r_s)^2$$
> At the token level, the error from this method arises from random noise.  Assuming we can whiten the rewards to have a mean of 0 and variance σ², the token-level error is:
>
> $$\text{err}_{\text{token level}} = \sum {c_i}^2 = \sigma^2N$$
>
> where $c_i$ represents the random noise, and $N$ is the total sequence length. The total error is the sum of sequence-level and token-level errors:
>
> $$\text{err} = err_s + err_t = \sum_{k=1}^{K} \sum_{t \in S_k} (r_t - r_k)^2 + \sigma^2 K$$
>
> where $K$ is the number of steps/sequences, and $S_k$ represents the set of tokens in the k-th step/sequence.  $r_k$ represents the reward assigned to the k-th sequence.
> This formula effectively illustrates two key issues:
>
> 1. Error: Steps must be effectively divided based on crucial information within the context to minimize the overall error across all tokens.
>
> 2. Variance in Token-Level Rewards: Why the reward signal at the token level exhibits greater variance.
>
> > **W2**: Questionable Effectiveness.
>
> Your suggestion provided a valuable alternative perspective, which led us to add Section 3.2.1 and appendix D.
>
> We would like to clarify that the mask used here is a loss mask (as opposed to an attention mask). This technique is widely adopted in the community, for instance, in the HuggingFace TRL framework and others' OpenRLHF frameworks.
>
> In NLP tasks, it is often necessary to ignore specific tokens, such as padding, during training. Here is a detailed explanation of how masking works with cross-entropy loss:
>
> **Cross-Entropy Loss Definition**:
> $$L = -\sum_{i} y_i \log(p_i) $$
>
> **Applying Mask to the Loss**:
>       Define a mask $ m_i$, where $ m_i = 0 $ if the token at position $ i $ is to be ignored, and $m_i = 1$ if it should be included in the loss.
> To ignore tokens, the masked loss is calculated as:
> $$ L = -\sum_{i} m_i \, y_i \log(p_i)$$
>   This ensures that positions where $m_i = 0$ contribute zero to the loss, effectively ignoring those tokens.
>
> **Effect on Gradients**:
> $$
>    m_i \, y_i \log(p_i) = 0 \quad \text{if} \quad m_i = 0
> $$
>
> This approach allows for selective backpropagation, ensuring that only relevant tokens influence the model's parameter updates.
>
> You can refer to the following works：
> * OpenRLHF: https://github.com/OpenRLHF/OpenRLHF/blob/78e1fbb7f34cb313fe63cc0eb0a6ba5b7ed764a9/openrlhf/trainer/dpo_trainer.py#L388
> * TRL: https://github.com/huggingface/trl/blob/066fc37bd3381e5de3ac0bb988ce834a050c459f/trl/trainer/dpo_trainer.py#L1150
>
> > **W3-4&W6**: Poor Writing Quality.
>
> We truly apologize for this issue.
> The literature review has been significantly condensed, focusing on a comparison of the most relevant work. Section 5.1 now covers alignment（As reviewer S1Kp'opinion）and RLHF, while Section 5.2 discusses research on fine-grained reward signals.
> We have rechecked the formatting of the entire paper. Improvements include a comprehensive check and correction of spacing, formulas, and citations.

---

> ### Author Response · Authors · 2024-12-02
> **Response to Reviewer BZ8y [Part 2]**
>
> > **Q1**: explain the concept of the gap between action and reward space alignment.
>
> In autoregressive generative transformers, each token represents an action. The size of the vocabulary is the size of the action space. As mentioned in Section 1, in RLHF the action space is typically more sparse than the reward space. Therefore, we need to increase the density of the reward space to align them.
>
> > **Q2**: How is the threshold b in Eq. 11 determined
>
> We have two methods for setting the b.
>
> 1. Offline sampling to calculate the mean reward score, and then using 0 as the baseline after subtracting this mean from the reward model's scores.
>
> 2. The second method starts by using 0 as the baseline to distinguish positive and negative rewards at step 0 of the experiment. Simultaneously, we initialize an array to store the current step and the mean reward. As training progresses, we dynamically update this mean based on the reward scores given by the reward model, and use this updated mean as the baseline.
>
> In our limited experiments, we found the first method to be more effective. All experiments in this paper utilize the first method.
>
> > **Q3&W5**: Have you experimented with alternative model architectures?
>
> Yes. We've observed similar results on both Llama 3 and our closed-source model. If you believe it's absolutely necessary, I can supplement with experiments on Llama 3.
>
> > **Q4**: What are the additional computational costs associated with this method?
>
> It requires fine-grained annotations (compared to the sentence-level preference annotations in RLHF), or a well-trained reward model.

---

### Official Review · Reviewer_wBaN · 2024-11-03

**Soundness:** 2
**Presentation:** 1
**Contribution:** 2
**Rating:** 3
**Confidence:** 4

**Summary:**

The paper presents idea of assigning scalar rewards to subsequences of LLM-generated text. The main idea behind their methodology is to find a mask over the text, a mask to hide negative aspects of a winning answer and positive aspects of a losing answer in a preference learning setting. Then use the masked sequences for doing Reinforcement Learning from Human Feedback (RLHF) with (1) standard reward modeling based RLHF, (2) implicit RLHF with direct preference optimization (DPO) and (3) rejection sampling variant of RLHF. Token-level masks are generated by using a reference value  $b$ and a neutral region parameter $\delta$, if the token's reward is more than $b + \delta$ or is less than $b-\delta$, accordingly +1 or -1 mask is generated, otherwise is set to zero. Since the token level rewards are different for different sentences, mask gets adapted per sentence. The paper then applies masked variants of RLHF algorithms for tasks like language generation, reasoning, code, etc. The results demonstrate incremental improvements on various benchmarks.

**Strengths:**

The issues behind the sentence-level reward assignment and token-level reward assignment in RLHF are pointed out aptly by the paper: sentence-level being too coarse while token-level being prone to high variance. Thus, the paper's investigation on finding appropriate message-level techniques for reward assignment is highly relevant.

The strength of the technique is in its simplicity. Idea of using masks for inducing better reward assignments to token-level RLHF in order to reduce the variance is neat. Being simple, the technique finds easy pluggable application to different RLHF paradigms.

The experimental setup tests the idea on multiple benchmarks, and the proposed algorithm shows improvements across most of them. I am unsure about the significance of these improvements as they come off rather incremental, except for the Human Eval dataset.

**Weaknesses:**

Lack of depth in the proposed algorithm: Although simple, the technique used in the paper lacks depth. Schmitt trigger approach used in the paper is not referenced properly and hence very hard to assign any significance to the particular idea. Masking with neutral region is a straightforward rule and does not need any control theory unless the particular ablations are shown on other masking techniques.

Statistical significance of the results: The plots in the paper do not use multiple random seeds, this makes it hard to gauge the effectiveness of the technique. Further, the table 1 of the main results does not show any mean and standard-deviation analysis of the values, and the statistical superiority of the results cannot be confirmed.

The obscurity in the writing of the paper:
- Lines 46-48: Not clear what is meant by sample level supervision, and why it leads to any inconsistency between rewards and action spaces.
- In multiple places, new sentence starts just after a period.
- ORM and PRM terms are used, again, without any introduction for these terms. Also, the whole introduction section seems loosely assembled without making any logical flow. I had to understand the paper by going through entire text and then seeing things in the context of paragraphs appearing later in the paper.
- What is the purpose behind figure 2? What do the colors used on the text block mean? Also, what is the conclusion of the plot on the right? Why does the plot have two dotted yellow boxes?
- Figure 3 requires additional explanations -- clarifying why not mask both the positive and the negative aspects of any answer with +1/-1. The paper currently proposes only to mask the negative aspect in a positive sentence and positive aspect of a negative sentence, but then in equation 16, proposes a tertiary masking instead of binary.
- Figure 4, I am unsure how to read this figure to understand Schmitt triggers.
- What is difference between equation 6 and equation 8? Aren't both showing the same thing unless $y_{win}$ is defined separately?

**Questions:**

Questions:

- What are the authors referring to by "understanding gap between action and reward space in alignment" in the title?

- "Human Eval et al." do you mean "Human Eval etc."?

Additional suggestions:
- A thorough pass through the introduction section would be highly beneficial for this paper.
- Section 2.3 on contextual dueling bandits is never used in any methodological development in the paper and thus can be removed from the main paper.

- Use of multiple seeds for the experiments would be highly recommended as it will give statistical strength to the results demonstrated.

- Also, figures 6 and 7 would benefit by labeling the axes properly. The plots have very small legends, do not have any information on x or y axes about what is to be understood.

- Please add control theory related references in the related section and a discussion on importance of Schmitt triggers or ablations on masking techniques which show the one used in the paper is superior. Also, I am not convinced about author's claims of using control theory in this paper as it is not clearly supported in theory, experiments or references.

- Please rewrite the equations in the preliminary section, the distribution over which expectation is taken are not properly formatted.


Overall, I find that the results of this paper are its main strength. The writing of this paper needs significant changes especially the introduction needs to be more lucid. Also, other aspects of formatting can be paid attention to. I am convinced to reject the current draft of this idea.

---

> ### Author Response · Authors · 2024-12-02
> **Response to Reviewer wBaN**
>
> We thank the reviewer wBaN for insightful comments and constructive suggestions, which have significantly improved the quality of this manuscript.
> > **W1**: Lack of depth in the proposed algorithm:
>
> Thank you for bringing this matter to our attention.
>
> * The introduction now contains the research objectives, significance, and main contributions of this work. Parts that duplicated the literature review have been removed or relocated to Section 5.
> * To provide a clearer problem definition, we've added Section 2.3. In Section 2.3, we provide a detailed definition of the problem. In Section 2.3, we define and differentiate between the two types of alignment discussed in this paper. We clarify that the alignment referred to in the title is preference alignment (Section 2.3.1) and also explain representational alignment (Section 2.3.2, noting the need for a denser reward signal to align the action space).
> * In Section 3.1, we identify potential sources of error when the reward signal is sparse, clarifying why our approach of identifying key information within the sequence can effectively divide subsequence and reduce errors.
>
>
> In Section 3.1, we define the supervisory signal error. Traditional methods assign the reward of a step to all tokens within that step/sequence. The error in this method stems from the discrepancy between the overall reward for the sequence $r_s$ and the reward assigned to individual actions/tokens $r_t$. The error is formulated as:$$\text{err}_{\text{sequence level}} = \sum (r_t - r_s)^2$$
> At the token level, the error from this method arises from random noise.  Assuming we can whiten the rewards to have a mean of 0 and variance σ², the token-level error is:
>
> $$\text{err}_{\text{token level}} = \sum {c_i}^2 = \sigma^2N$$
>
> where $c_i$ represents the random noise, and $N$ is the total sequence length. The total error is the sum of sequence-level and token-level errors:
>
> $$\text{err} = err_s + err_t = \sum_{k=1}^{K} \sum_{t \in S_k} (r_t - r_k)^2 + \sigma^2 K$$
>
> where $K$ is the number of steps/sequences, and $S_k$ represents the set of tokens in the k-th step/sequence.  $r_k$ represents the reward assigned to the k-th sequence.
> This formula effectively illustrates two key issues:
>
> 1. Error: Steps must be effectively divided based on crucial information within the context to minimize the overall error across all tokens.
>
> 2. Variance in Token-Level Rewards: Why the reward signal at the token level exhibits greater variance.
>
> > **W2**: Statistical significance of the results
>
> Thank you for raising this concern. While the robustness of the experiments could indeed be further improved, we believe this does not invalidate the core contributions and the correctness of the proposed approach. As other reviewers mentioned, extensive experimentation and testing may provide similar validation to running multiple experiments with different random seeds.
>
> > **W3&Additional suggestions**: The obscurity in the writing of the paper
>
> * sample level supervision: The terminology in this paper has been standardized: all instances of "sample level" have been replaced with "sequence level (wise)."
> * We have added content related to Process-supervised Reward Models in Section 2.1.
> * We've enhanced the figures with more detailed explanations and ensured that each figure is referenced within the main text.
> * The part of the Schmitt trigger has been moved to the experimental settings section and the appendix, and corresponding citations have been added.
> * We have rechecked the formatting of the entire paper, including a comprehensive check and correction of spacing, formulas, and citations.
> * The introduction now contains the research objectives, significance, and main contributions of this work. Parts that duplicated the literature review have been removed or relocated to Section 5.
> * Section 2.3 on contextual dueling bandits: This section, which discussed the challenges of sparse reward signals requiring adaptive credit assignment, has been removed for conciseness and practicality.
>
> > **Q1**: What are the authors referring to by "understanding gap between action and reward space in alignment" in the title?
>
> In autoregressive generative transformers, each token represents an action. The size of the vocabulary is the size of the action space. As mentioned in Section 1, in RLHF the action space is typically more sparse than the reward space. Therefore, we need to increase the density of the reward space to align them.
>
> > **Q2**: "Human Eval et al." do you mean "Human Eval etc."?
>
> Yes. Thank you for your suggestion. We have made this revision.

---

### Official Review · Reviewer_GTLV · 2024-11-03

**Soundness:** 2
**Presentation:** 1
**Contribution:** 2
**Rating:** 3
**Confidence:** 3

**Summary:**

This paper presents "Adaptive Message-wise RLHF," a new method to improve Reinforcement Learning from Human Feedback (RLHF) by focusing on specific parts of a sequence rather than an overall reward. Traditional RLHF can be inefficient because it doesn’t recognize which tokens should be emphasized or downplayed. The new method identifies key “pivot tokens” to adaptively target important information, breaking supervision down to a fine-grained, subsequence level. This approach works well across different training methods, like PPO, DPO, and rejection sampling, and adapts to a variety of tasks. Experiments show it reduces issues like hallucinations and forgetting, achieving a 10% improvement on adversarial samples and 1.6% on benchmarks like MMLU and GSM8K. Future plans include refining subsequence handling and exploring control theory to strengthen this framework.

**Strengths:**

- The proposed method is conceptually simple, intuitive, and easy to implement.

- The idea of introducing a Schmitt trigger to ensure the stability of the reward signal is innovative.

- The model’s performance was thoroughly validated through large-scale experiments, considering various benchmarks, evaluation metrics, and training strategies.

**Weaknesses:**

- **Attention to writing quality is needed:**
  - There are numerous instances where spacing is incorrect.
    - For example, in the caption for Figure 1, there appear to be missing spaces after periods or commas.
    - In Section 4.1, there is a missing space after the paragraph title as well.

- **The paper could benefit from separate sections for literature review and problem definition.** Currently, both are included in the introduction section, resulting in insufficient review and imprecise problem definitions.
  - This study aims to align the action space with the reward space by putting selective attention on useful tokens. Accordingly, I recommend the authors provide a formal and explicit definition of alignment and derive how token selection could improve this alignment.
  - Regarding the formal definition of "alignment," reviewing the following literature may provide some useful insights:
    - Wang, T., & Isola, P. (2020, November). *Understanding contrastive representation learning through alignment and uniformity on the hypersphere*. In International conference on machine learning (pp. 9929-9939). PMLR.
    - Wang, C., Yu, Y., Ma, W., Zhang, M., Chen, C., Liu, Y., & Ma, S. (2022, August). *Towards representation alignment and uniformity in collaborative filtering*. In Proceedings of the 28th ACM SIGKDD conference on knowledge discovery and data mining (pp. 1816-1825).

- **Consistent errors in subscript formatting were observed in equations.**
  - For example, in equations (4), (6), (7), (8), and (9) on page 4, subscripts are consistently misformatted. Additionally, a thorough review for typos in other parts of the manuscript is recommended before submission.

- **The theoretical foundation appears weak.** The strengths of the proposed masking-based method compared to sequence-level methods (e.g., learning efficiency, nuanced and coherent generation) require rigorous explanation and justification.
  - The completeness of the paper would be enhanced if mathematical proofs were provided to demonstrate how, compared to sequence-level methods, introducing Eq. (16) reduces the variance of the reward signal, thereby preventing reward over-optimization.
  - For example, let’s say $\mathbf{r} \in \mathbb{R}$ is a sequence-level reward while $r_{t} \in \mathbb{R}$ is a token-level reward at time step $t$. Given $\mathbf{r} = \frac{1}{T}\sum_{t=1}^{T}r_{t}$, showing $\text{Var}(\mathbf{r}) = \mathbb{E}[\mathbf{r} - \mathbb{E}[\mathbf{r}]]^{2} > \mathbb{E}\_{\substack{ (s, a) \sim \pi\_{\theta} \\\ r=R(s, a) \\\ \hat{r}=r\cdot M(s,a) } }[\hat{r}\_{t} - \mathbb{E}[\hat{r}_{t}] ]^{2}$ would be one possible approach.

- **Visual content preparation is lacking.**
  - For example, Figure 2 on page 2 has significant wasted space. Adjusting the size of the figure titled "Dense Reward Comparison" so that the top and bottom lines align with those of the prompt box on the left, and increasing the font size in the legend, would help eliminate unnecessary white space.
  - Additionally, the text in the legends of Figures 5, 6, and 7 is very small, making it difficult to read. Increasing the font size will enhance readability.

**Questions:**

- On line 157, page 3, the statement "$\textit{``This approach not only enhances learning efficiency but also potentially leads to more coherent and contextually appropriate language generation."}$" is made. What theoretical or empirical evidence supports this claim?

- On page 5, it is stated that "$\textit{``token-level optimization enables more nuanced preference capture and potentially more coherent outputs"}$." What theoretical or empirical evidence supports this claim?

- On line 300, page 6, "$\textit{dynamically updates the threshold}$" is mentioned. On line 313, this threshold appears to refer to the baseline value $b$. However, there is no explanation in the text on how this $b$ is adaptively updated. Could you clarify the update process?

- Why were the PRM800K and Helpsteer datasets selected? Could you explain the reason why only those datasets were used despite there exist well-known benchmark datasets for RLHF studies, e.g., Anthrophic-3H dataset.

---

> ### Author Response · Authors · 2024-12-02
> **Response to Reviewer GTLV**
>
> We sincerely thank Reviewer GTLV for providing the insightful review, and we have revised our paper based on your review carefully.
> > **W1&W3: Attention to writing quality is needed & Consistent errors in subscript formatting were observed in equations.**
>
> We truly apologize for this issue. We have rechecked the formatting of the entire paper. Improvements include a comprehensive check and correction of spacing, formulas, and citations.
>
> > **W2: The paper could benefit from separate sections for literature review and problem definition.**
>
> Thank you for your insightful suggestion.
> * The introduction now contains the research objectives, significance, and main contributions of this work. Parts that duplicated the literature review have been removed or relocated to Section 5.
> * To provide a clearer problem definition, we've added Section 2.3. In Section 2.3, we provide a detailed definition of the problem. In Section 2.3, we define and differentiate between the two types of alignment discussed in this paper. We clarify that the alignment referred to in the title is preference alignment (Section 2.3.1) and also explain representational alignment (Section 2.3.2, noting the need for a denser reward signal to align the action space).
> * In Section 3.1, we identify potential sources of error when the reward signal is sparse, clarifying why our approach of identifying key information within the sequence can effectively divide subsequence and reduce errors.
>
> > **W3: The theoretical foundation appears weak.**
>
> We appreciate your insightful suggestion and have enhanced the theoretical analysis section.
>
>  We've added Section 3.2, which provides a problem definition, and Section 3.1, which theoretically derives the optimal solutions for error and variance.
> In Section 3.1, we define the supervisory signal error. Traditional methods assign the reward of a step to all tokens within that step/sequence. The error in this method stems from the discrepancy between the overall reward for the sequence $r_s$ and the reward assigned to individual actions/tokens $r_t$. The error is formulated as:$$\text{err}_{\text{sequence level}} = \sum (r_t - r_s)^2$$
> At the token level, the error from this method arises from random noise.  Assuming we can whiten the rewards to have a mean of 0 and variance σ², the token-level error is:
>
> $$\text{err}_{\text{token level}} = \sum {c_i}^2 = \sigma^2N$$
>
> where $c_i$ represents the random noise, and $N$ is the total sequence length. The total error is the sum of sequence-level and token-level errors:
>
> $$\text{err} = err_s + err_t = \sum_{k=1}^{K} \sum_{t \in S_k} (r_t - r_k)^2 + \sigma^2 K$$
>
> where $K$ is the number of steps/sequences, and $S_k$ represents the set of tokens in the k-th step/sequence.  $r_k$ represents the reward assigned to the k-th sequence.
> This formula effectively illustrates two key issues:
>
> 1. Error: Steps must be effectively divided based on crucial information within the context to minimize the overall error across all tokens.
>
> 2. Variance in Token-Level Rewards: Why the reward signal at the token level exhibits greater variance.
>
> > **W4: Visual content preparation is lacking.**
>
> We've adjusted the size of the images and legends.
>
> > **Q1-2: A claim about dense supervision signals improve learning efficiency and generation quality.**
>
> Better allocation of reward signals can improve efficiency and training outcomes compared to relying on the model to perform adaptive credit assignment on tasks where it struggles. This is widely acknowledged within the RL community, for example:
> * Andrychowicz, Marcin, et al. "Hindsight experience replay." Advances in neural information processing systems 30 (2017).
>
> But your question is excellent. This paper doesn't offer a theoretical analysis or experimental results on this point, so these statements have been removed.
>
> > **Q3: About dynamically updating the baseline.**
>
> We have two methods for setting the baseline.
>
> 1. Offline sampling to calculate the mean reward score, and then using 0 as the baseline after subtracting this mean from the reward model's scores.
>
> 2. The second method starts by using 0 as the baseline to distinguish positive and negative rewards at step 0 of the experiment. Simultaneously, we initialize an array to store the current step and the mean reward. As training progresses, we dynamically update this mean based on the reward scores given by the reward model, and use this updated mean as the baseline.
>
> In our experiments, we found the first method to be more effective. All experiments in this paper utilize the first method.
>
> > **Q4: selection on datasets**
> 1. HelpSteer's richer data domain, encompassing longer sequences, makes it better suited to our needs than other datasets. The inclusion of PRM800K further ensures comprehensive coverage across a diverse range of tasks.
>
> 2. Our annotation resources are limited. Step-level annotation of the dataset is prohibitively expensive.

---

> ### Comment · Reviewer_GTLV · 2024-12-03
> **Response to the rebuttal.**
>
> Dear Authors,
>
> Thank you for your efforts in addressing my concerns.
>
> I’m glad to see that W1, W3, and W5 have been successfully resolved. However, W2 and W4 still require further attention.
>
> **Problem Definition:**
> It would be helpful to present the problem definition in a more formalized manner, as is standard in the ML community. A formalized approach would make the concept of 'alignment' more mathematically precise and persuasive for readers. Unfortunately, the revised version still lacks this level of clarity.
>
> **Section 3.1:**
> This section still falls short in providing theoretical insights into the masking-based approach. While it explains how to quantify the mismatching error between token-level rewards and sequence-level rewards, it does not offer a theoretical explanation of how masking certain subsequences can help reduce this error. For example, it is well known that masking-based loss function can be formalized as a composite likelihood function, and therefore, mathematically showing that minimization of the composite (or pseudo) likelihood objective (in the setting of interest) leads to the variance reduction in rewards, compared to the general likelihood function, can be one promising approach.
>
> Given these points, I will keep my score unchanged.
>
> Once again, I sincerely appreciate the authors’ efforts in addressing my concerns.

---

> ### Author Response · Authors · 2024-12-03
>
> Thank you again for your patient review and guidance.
> > Problem Definition
>
> The term "alignment" in the title refers to aligning model outputs with human preferences, a notion we believe is generally familiar to readers. For example:
> * Tang, Yunhao, et al. "Understanding the performance gap between online and offline alignment algorithms." arXiv preprint arXiv:2405.08448 (2024).
> * Guo, Shangmin, et al. "Direct language model alignment from online ai feedback." arXiv preprint arXiv:2402.04792 (2024).
> * Xu, Shusheng, et al. "Is dpo superior to ppo for llm alignment? a comprehensive study." arXiv preprint arXiv:2404.10719 (2024).
> * Zhu, Banghua, et al. "Fine-tuning language models with advantage-induced policy alignment." arXiv preprint arXiv:2306.02231 (2023).
>
> > Section 3.1
>
> This section simply identifies a source of error（from reward model）. A detailed explanation and derivation of the masking method are provided in Section 3.2.1 and Appendix D.
>
> This technique is widely adopted in the community, for instance, in the HuggingFace TRL and OpenRLHF.
>
> In NLP tasks, it is often necessary to ignore specific tokens, such as padding (<pad>), during training. Here is a detailed explanation of how masking works with cross-entropy loss:
>
> **Cross-Entropy Loss Definition**:
> $$L = -\sum_{i} y_i \log(p_i) $$
>
> **Applying Mask to the Loss**:
>       Define a mask $ m_i$, where $ m_i = 0 $ if the token at position $ i $ is to be ignored, and $m_i = 1$ if it should be included in the loss.
> To ignore tokens, the masked loss is calculated as:
> $$ L = -\sum_{i} m_i \, y_i \log(p_i)$$
>   This ensures that positions where $m_i = 0$ contribute zero to the loss, effectively ignoring those tokens.
>
> **Effect on Gradients**:
> $$
>    m_i \, y_i \log(p_i) = 0 \quad \text{if} \quad m_i = 0
> $$
>
> This approach allows for selective backpropagation, ensuring that only relevant tokens influence the model's parameter updates.
>
> You can refer to the following works：
> * OpenRLHF: https://github.com/OpenRLHF/OpenRLHF/blob/78e1fbb7f34cb313fe63cc0eb0a6ba5b7ed764a9/openrlhf/trainer/dpo_trainer.py#L388
> * TRL: https://github.com/huggingface/trl/blob/066fc37bd3381e5de3ac0bb988ce834a050c459f/trl/trainer/dpo_trainer.py#L1150
>
> The masking mechanism is not the key point. Our work here uses it to explore the potential benefits of segmenting steps based on rewards from the reward model, suggesting that this approach may be more effective than using punctuation like periods or semicolons.

---

### Note · Authors · 2024-12-05

I have read and agree with the venue's withdrawal policy on behalf of myself and my co-authors.